# Rapid Plug-in Defenders

**Kai Wu**
Xidian University
kwu@xidian.edu.cn

**Yujian Betterest Li**[*]
Xidian University
bebetterest@outlook.com

**Jian Lou**
Zhejiang University
jian.lou@hoiying.net

**Xiaoyu Zhang**
Xidian University
xiaoyuzhang@xidian.edu.cn

**Handing Wang**
Xidian University
hdwang@xidian.edu.cn

**Jing Liu**
Xidian University
neouma@mail.xidian.edu.cn

## Abstract

In the realm of daily services, the deployment of deep neural networks underscores the paramount importance of their reliability. However, the vulnerability of these networks to adversarial attacks, primarily evasion-based, poses a concerning threat to their functionality. Common methods for enhancing robustness involve heavy adversarial training or leveraging learned knowledge from clean data, both necessitating substantial computational resources. This inherent time-intensive nature severely limits the agility of large foundational models to swiftly counter adversarial perturbations. To address this challenge, this paper focuses on the **Ra**pid **P**lug-in **D**efender (**RaPiD**) problem, aiming to rapidly counter adversarial perturbations without altering the deployed model. Drawing inspiration from the generalization and the universal computation ability of pre-trained transformer models, we propose a novel method termed **CeTaD** (**C**onsidering Pr**e**-trained **T**ransformers **a**s **D**efenders) for RaPiD, optimized for efficient computation. **CeTaD** strategically fine-tunes the normalization layer parameters within the defender using a limited set of clean and adversarial examples. Our evaluation centers on assessing **CeTaD**'s effectiveness, transferability, and the impact of different components in scenarios involving one-shot adversarial examples. The proposed method is capable of rapidly adapting to various attacks and different application scenarios without altering the target model and clean training data. We also explore the influence of varying training data conditions on **CeTaD**'s performance. Notably, **CeTaD** exhibits adaptability across differentiable service models and proves the potential of continuous learning.

## 1 Introduction

It has been observed that trained neural network models exhibit vulnerability, failing to correctly predict labels when slight perturbations are added to the input examples [15, 2, 5]. This method, known as an evasion-based adversarial attack, poses a significant challenge. Recent research works [47, 42, 36, 41, 31] have focused on developing robust models by leveraging clean data knowledge or employing adversarial training techniques.

---

[*]Corresponding author

38th Conference on Neural Information Processing Systems (NeurIPS 2024).

Presently, deep neural networks serve as fundamental components across various domains [25, 12]. The most prominent models are pre-trained transformers, such as GPT-2 [33], BERT [9], and VIT [11]. Following pre-training on relevant data, they demonstrate strong generalization capabilities and can swiftly adapt to downstream tasks.

Defending deployed service models presents a more challenging scenario. These service models may face difficulties in fine-tuning when under attack, especially if methods like pruning [52] were implemented before deployment to compress or accelerate the service. Retraining a more robust model incurs a considerable computational cost and time. Furthermore, swift defense becomes crucial to prevent further losses instead of waiting for adversarial training or redeploying the model.

Hence, **Ra**pid **P**lug-**in D**efender (**RaPiD**), the goal of which is to swiftly counter adversarial perturbations in different application scenarios without altering the deployed model, is of much importance. Most current methods, especially non-invasive defenders (Detection: Magnet[28], ADN[29], DG[1], etc; Purification: HGD[23], Defense-GAN[35], CAFD[50], DISCO[17], DensePure[43], etc.) fail to accomplish this task due to the following reasons: First, they cannot rapidly defend with limited data due to heavy training with much time and training data. Especially, DM-Improves-AT [42] applies adversarial training on a large amount of data generated by diffusion models. Second, they cannot quickly adapt to different application scenarios. For example, as shown in Table 3, on CIFAR-10, although DiffPure [31] trained on CIFAR-10 (DiffPure$_{\textbf{CIFAR-10}}$) performs well against StAdvAttack, it could hardly work with that trained on Imagenet-1k (DiffPure$_{Imagenet-1k}$). Training a diffusion model on a specific field needs much time and data.

Table 1: Comparison of conditions between **RaPiD** and related works in adversarial defense, concentrating on requirements such as generating additional data, target service model tuning, heavy adversarial training application, utilization of clean data information, and the plug-in nature of the defense.

| Case | Data Generation | Tuning Service | Heavy Adversarial Training | Clean Data | Plug-in |
|---|---|---|---|---|---|
| DM-Improves-AT [42] | ✓ | ✓ | ✓ | ✓ | ✗ |
| DyART [47] | ✗ | ✓ | ✓ | ✓ | ✗ |
| FD [46] | ✗ | ✗ | ✓ | ✓ | ✓ |
| DISCO [17] | ✗ | ✗ | ✓ | ✓ | ✓ |
| GDMP [41] | ✗ | ✗ | ✗ | ✓ | ✓ |
| DiffPure [31] | ✗ | ✗ | ✗ | ✓ | ✓ |
| DensePure [43] | ✗ | ✗ | ✗ | ✓ | ✓ |
| R&P [45] | ✗ | ✗ | ✗ | ✗ | ✓ |
| CeTaD (Ours) | ✗ | ✗ | ✗ | ✗ | ✓ |

In this case, we find that the large volume of training data and the substantial number of parameters requiring adjustment are the primary culprits causing the time-consuming nature of current methods. Motivated by the generalization and the universal computation ability of pre-trained transformer models [26, 19], as well as evidence that pre-training can fortify robustness [16], we propose a new defense method, **CeTaD**—**C**onsidering **P**re-trained **T**ransformers **a**s **D**efenders. **CeTaD** is a plug-in defender, which initializes by pre-trained weights and fine-tunes minimal parameters with only few-shot samples. Notably, **CeTaD** diverges from existing methods as follows: It demonstrates efficacy with a minimal sample size required for fine-tuning the rapid defender. Additionally, **CeTaD** avoids the need for modification within the deployed model, ensuring adaptability across diverse application scenarios, especially with large foundational models.

Our experimental results demonstrate that, in the context of **RaPiD**, **CeTaD** exhibits superior performance concerning both clean accuracy and adversarial accuracy with limited training data and computational resources, compared to feasible baselines. The method's effectiveness spans across various datasets and attack methods. The minimal tuned parameters mitigate the risk of overfitting during the training process. Through ablation studies, we evaluate the components within **CeTaD**, such as the residual connection and parameter initialization. Furthermore, we explore the impact of

data scale and balance, while the transfer test underscores its potential for generalization, with the transfer gap potentially bolstering robustness. Our contributions are summarized as follows.

1. Introduction of the Rapid Plug-in Defender framework aimed at promptly addressing adversarial perturbations quickly without altering the deployed model.

2. Utilization of two strategies in **RaPiD** to expedite response time: a) leveraging Pretrained models to minimize parameter updates, b) employing few-shot samples for defender training.

3. **CeTaD**'s achievement of defense response within half an hour on a single GPU, surpassing the current **RaPiD** method in efficacy. Additionally, **CeTaD** demonstrates proficiency in defending against diverse attacks and enables zero-shot transfer to different datasets.

## 2 Related Work

**Adversarial Examples and Defenses**. Introduced by [38], adversarial examples could fool a neural network into working incorrectly. Among various methods [2, 5], attacks in a white-box manner are usually the most dangerous since the leaked information of the victim model is utilized. Many efforts generate adversarial examples through gradients of victims. [15] yielded a simple and fast method of generating adversarial examples (FGSM). [4] proposed much more effective attacks tailored to three distance metrics. PGD is a multi-step FGSM with the maximum distortion limitation [27]. [8] came up with AutoAttack, a parameter-free ensemble of attacks. Facing adversarial examples, lots of effort pay attention to defense [7]. Some works detect adversarial attack in advance. [28] learned to differentiate between normal and adversarial examples by approximating the manifold of normal examples. [29] proposed to augment deep neural networks with a small detector subnetwork which is trained on the binary classification task of distinguishing genuine data from data containing adversarial perturbations. [1] constructed a Latent Neighborhood Graph for detection. Some works strengthen robustness by adversarial training, where the model would be trained on adversarial examples [15]. [42] proposed to exploit diffusion models to generate much extra data for adversarial training. [47] encouraged the decision boundary to engage in movement that prioritizes increasing smaller margins. In addition, many works focus on adversarial purification. [23] proposed high-level representation guided denoiser for purification. [35] trained a generative adversarial network to model the distribution of unperturbed images. [50] proposed to remove adversarial noise by implementing a self-supervised adversarial training mechanism in a class activation feature space. [36] combined canonical supervised learning with self-supervised representation learning to purify adversarial examples at test time. [17] purified adversarial examples by localized manifold projections. Similar to [41], [31] followed a forward diffusion process to add noise and recover the clean examples through a reverse generative process. Furthermore, [43] consisted of multiple runs of reverse process for multiple reversed samples, which are then passed through the classifier, followed by majority voting of inferred labels to make the final prediction.

**Few-shot Adversarial Training**. Here are several work on adversarial training with few-shot samples. Many works focus on few-shot learning by adversarial training. [30] applied adversarial discriminator for supervised adaptation problem. [49] introduced an adversarial generator to help few-shot models learn sharper decision boundary. [22] proposed to hallucinate diverse and discriminative features on few labeled samples. Furthermore, few-shot adversarial training is utilized to enhance the robustness. [13] developed an adversarial training algorithm for producing robust meta-learners and found the the meta-learning models are the most robust with only the last layer tuned. [10] integrated a adversarial-aware classifier, adversarial-reweighted training and a feature purifier. In this paper, we implement fine-tuning with few-shot adversarial examples for swift defense response.

**Pre-trained Transformer**. Introduced by [40], transformer is an efficient network architecture based solely on attention mechanisms. It is first applied in natural language processing and then rapidly spread in computer vision. [9] proposed BERT to utilize only the encoder of transformer while GPT-2 [33] considered only transformer decoder. In computer vision, [11] proposed Vision Transformer (VIT), transforming a image into sequences of patches and processing them through a pure encoder-only transformer. Moreover, transformer has the ability of universal computation over single modality. [26] demonstrated transformer models pre-trained on natural language could be transferred to tasks of other modalities. Similar to [51, 48], [39] proposed to make the frozen language transformer perceive images by only training a vision encoder as the sequence embedding.

To strengthen robustness and generalization, we initialize the plug-in defender by a language/vision pre-trained transformer model and only fine-tune minimal parameters.

## 3  Pre-trained Transformers as Defenders

**Definition 1 RaPiD (Rapid Plug-in Defender):** *RaPiD is a defense mechanism in machine learning designed to swiftly mitigate adversarial perturbations encountered by deployed models without necessitating alterations to the model's architecture or parameters. Its primary objective is to provide rapid and effective protection against adversarial attacks while maintaining the integrity and functionality of the existing deployed model.*

For **RaPiD** implementation, the victim service model $\mathbf{M}$ is fixed, with limited clean data $\mathbf{X_c}$ and a sparse set of potentially imbalanced adversarial examples $\mathbf{X_a}$ from a single attack method for training, all labeled under $\mathbf{Y}^*$. While this paper specifically addresses image classification within the service task, the approach holds theoretical promise for other tasks as well. By default, only one-shot imbalanced adversarial examples are accessible.

**CeTaD**, initializes with pre-trained weights from models like VIT or BERT. It involves a defender embedding and decoder to align the plug-in defender with the input example and the service model respectively. A residual connection retains the primary input features, resulting in the original input combined with the defender's output serving as the input to the service model. Default settings include copying the embedding from VIT or BERT and utilizing PixelShuffle [37] for the decoder. Especially, PixelShuffle rearranges the elements, unfolding channels while increasing spatial resolution, to match the image resolution. Given limited access to adversarial examples, we opt to fine-tune minimal parameters, such as layer normalization, in the plug-in defender, mitigating overfitting and excessive bias on clean data.

The method is formulated as follows: In a single-label image classification task, each image $x_c$ from the clean set $\mathbf{X_c}$ is attached with a label $y^*$ within the corresponding label set $\mathbf{Y}^*$. A deployed model $\mathbf{M}$ maps $x_c$ into the prediction $y_c$ as $y_c = \mathbf{M}(x_c)$.

If the model $\mathbf{M}$ works correctly, $y_c = y^*$. Utilizing leaked information about $\mathbf{M}$, the attacker edits the original image $x_c$ to an adversarial image $x_a$ by introducing noises, resulting in an adversarial image $x_a$ within the adversarial set $\mathbf{X_a}$. The prediction for $x_a$ is then determined as

$$y_a = \mathbf{M}(x_a) \tag{1}$$

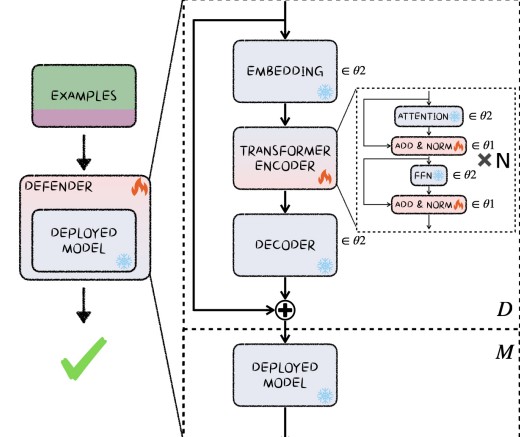

Figure 1: The structure of **CeTaD**. The input example would be added with the feature obtained by the stack of an embedding, a transformer encoder, and a decoder before being processed by the deployed service model. The deployed model is frozen in **RaPiD**.

If the attack succeeds, $y_a \neq y^*$. The tuning set for defense, denoted as $\mathbf{X_d}$, represents a subset of $\mathbf{X_a}$ and has a limited size within the **RaPiD** framework. Our approach incorporates a defender module $\mathbf{D}$ with parameters $\theta$, while maintaining $\mathbf{M}$ fixed. As illustrated in Fig. 1, $\mathbf{M}$ consists of the embedding of a pre-trained VIT, a pre-trained transformer encoder as a feature poccessor, and a parameter-free PixelShuffle block as a decoder. Only limited parameters are fine-tuned within $\mathbf{X_d}$. The loss function is formulated as

$$\arg\min_{\theta_1} \sum_{x_d \in \mathbf{X_d}} loss(\mathbf{M}(\mathbf{D}_{\theta_1,\theta_2}(x_d) + x_d), y^*) \tag{2}$$

where *loss* is the cross-entropy for classification. Within this framework, $\theta_1$ and $\theta_2$ represent the parameters of $\mathbf{D}$, with only $\theta_1$ being subject to tuning. Specifically, $\theta_1$ pertains to the layer normalization parameters, while $\theta_2$ encapsulates the remaining parameters. With the trained defender $\mathbf{D}_{\theta_1^*,\theta_2}$, the final prediction $y$ is obtained as

$$y = \mathbf{M}(\mathbf{D}_{\theta_1^*,\theta_2}(x') + x') \tag{3}$$

Table 2: Accuracy performance with different methods for **RaPiD**.

| Method | CA(%) | AA(%) |
|---|---|---|
| *None* | 93.75 | 00.00 |
| R&P ([45]) | 93.16 | 02.34 |
| RN(std=0.05) ([32]) | 68.95 | 05.86 |
| RN(std=0.06) ([32]) | 57.23 | 11.13 |
| RN(std=0.07) ([32]) | 48.24 | 13.67 |
| Linear | 23.44 | 21.68 |
| FFN | 18.95 | 19.34 |
| Bottleneck | 23.44 | 20.90 |
| FD ([46]) | 37.50 | 23.83 |
| **CeTaD**-GPT-2 | 55.08 | 39.65 |
| **CeTaD**-VIT | 82.81 | 30.27 |
| **CeTaD**-VIT-large | 71.68 | 44.14 |
| **CeTaD**-BERT | 68.75 | 44.34 |
| **CeTaD**-BERT-large | 66.02 | 48.83 |

Table 3: Accuracy performance with StAdvAttack on CIFAR-10.

| Method | CA(%) | AA(%) |
|---|---|---|
| *None* | 93.75 | 00.00 |
| R&P | 92.19 | 01.76 |
| RN(std=0.05) | 70.90 | 01.95 |
| RN(std=0.07) | 46.29 | 05.27 |
| RN(std=0.09) | 32.03 | 08.20 |
| Linear | 18.36 | 15.43 |
| FFN | 20.12 | 16.01 |
| Bottleneck | 18.35 | 13.47 |
| FD | 29.49 | 14.64 |
| DiffPure$_{\textbf{CIFAR-10}}$ | 87.50 | 71.88 |
| DiffPure$_{\text{Imagenet-1k}}$ | 91.41 | 00.59 |
| **CeTaD**-GPT-2 | 14.64 | 09.57 |
| **CeTaD**-VIT | 84.57 | 00.39 |
| **CeTaD**-VIT-large | 67.57 | 06.64 |
| **CeTaD**-BERT | 56.25 | 11.52 |
| **CeTaD**-BERT-large | 56.64 | 17.38 |

where $\theta_1^*$ is the optimized parameters, and $x' \in (\mathbf{X_c} \bigcup \mathbf{X_a})$.

**Module Selection**. Module selection is a critical aspect of **CeTaD** given the limited parameter tuning. The embedding and decoder modules play pivotal roles in facilitating the mapping between the input and hidden spaces. Meanwhile, the encoder holds paramount importance for discerning adversarial cues and fortifying robustness, as it remains the sole trainable module and undertakes the most computation within **CeTaD**. Flexibility characterizes **CeTaD**, contingent upon ensuring harmonious dimensions across the modules.

Outlined below are succinct introductions to the utilized modules: BERT [9], a transformer encoder model, pre-trained for masked language modeling (MLM) and next sentence prediction (NSP) on a substantial uncased English dataset (available in base and large versions); VIT [11], a transformer encoder model, pre-trained for image classification on ImageNet-21k at a resolution of 224x224 (accessible in base and large versions); GPT-2 [33], a transformer decoder model, pre-trained for causal language modeling (CLM) on an extensive English corpus (available in 124M variant); PixelShuffle [37], a technique reorganizing elements within channels to enhance spatial resolution.

**Details of Optimization**. In our default setup, only layer norm parameters (48 parameter groups, 36864 variables in total) are fine-tuned using Lion [6] with default hyper-parameters. We optimize Eq. (2) over 500 epochs with a batch size of 32.

**Discussion**. Two perspectives elucidate **CeTaD**'s functionality. Initially, it functions akin to a purifier, detecting and filtering adversarial perturbations by introducing adaptive noise. Alternatively, akin to prompt engineering in natural language processing [24], **CeTaD** can be perceived as a prompt generator, creating adaptive prompts. These prompts serve as cues for the service model, aiding in enhanced classification of adversarial examples.

## 4 Experiments

**Experimental Setup**    *Datasets*. Three image classification datasets, MNIST [21], CIFAR-10 [20], CIFAR-100 [20], and Imagenet-1k[34], are utilized. We utilize the library, *Datasets*, to prepare data. For simplicity, the training set only consists of adversarial examples whose number equals to that of the classes, namely one-shot. The detailed information of datasets and pretrained models can be found in Appendix Section A.

*Attacks*. Evasion methods PGD [27], AutoAttack [8] and StAdvAttack [44] simulate service model leakage. Following [42], $l_\infty$-norm's max distortion is 8/255 and $l_2$-norm is 128/255. PGD parameters include ten iterations and step size $\epsilon/4$. Metrics include Clean Accuracy (**CA**) and Adversarial

Accuracy (**AA**). Clean accuracy stands for the accuracy on clean data while adversarial accuracy on adversarial data.

*Pre-trained Models*. Reproducibility relies on public models and checkpoints available on GitHub or Huggingface. For MNIST, the victim model is a fine-tuned VIT-base; for CIFAR-10, both of a fine-tuned VIT-base and a standardly trained WideResNet-28-10 are considered as victims; for CIFAR-100, a fine-tuned VIT-base is the victim; for Imagenet-1k, VIT-base is the victim. Pre-trained BERT-base, BERT-large, VIT-base, VIT-large and GPT-2-124M are considered as the choices of the defender initialization.

*Other Details*. Default settings include BERT-base defending WideResNet-28-10 against $l_\infty$-PGD on CIFAR-10, with the defender's embedding sourced from pre-trained VIT.

*Baselines* R&P [45] and Random Noise [32] are training-free, while the others (Linear, FFN, Bottleneck and FD [46]) undergo optimization. R&P employs random resizing and padding against adversarial examples, while Random Noise adds zero-mean normal distribution noise similar to BaRT [32]. Linear, FFN, Bottleneck, and FD replace module **D** in Fig. 1, maintaining the rest akin to CeTaD. Linear uses a single layer sans activation, FFN has two doubled hidden feature linear layers with a RELU activation, Bottleneck halves the hidden feature dimension, and FD integrates non-local denoising with a 1x1 convolution and identity skip connection, performing optimally at a hidden dimension of 256. Our method distinguishes itself from prior adversarial training approaches like [42], excelling in rapid defense without extensive retraining needs.

**CeTaD versus Baselines**   We compare **CeTaD** with other possible structures and feasible state-of-the-art baselines for **RaPiD**. Here, BERT-base is the defender for the WideResNet-28-10 against $l_\infty$-PGD on CIFAR-10. The results are shown in Table 2.

R&P maintains clean accuracy but shows minimal improvement in adversarial accuracy. Adding random noise slightly boosts adversarial accuracy but drastically reduces clean accuracy. Generally, training-free methods perform worse in adversarial accuracy compared to optimized ones like Linear, FFN, and Bottleneck, which perform similarly. With the fixed denoising structure and limited tuned parameters, FD outperforms other prior methods shown.

However, **CeTaD**, initialized by GPT-2, VIT, VIT-large, BERT, or BERT-large, excels in adversarial accuracy while maintaining high clean accuracy compared to the aforementioned methods. Notably, GPT-2-based initialization shows relatively poor performance, suggesting the need for better fusion of information between preceding and subsequent patches in visual tasks. Scaling matters too, as larger-scale defenders outperform their base-scale counterparts in adversarial accuracy.

Table 4: Accuracy performance in zero-shot transfers from the top to the bottom. "Source" refers to the environment where the defender is tuned, while "Target" represents the environment to which the defender transfers. *None* denotes direct training of the defender in the target environment without transfer.

| Target Data (Target Model) | Defender | Source Data (Source Model) | CA(%) | AA(%) |
|---|---|---|---|---|
| CIFAR-10 (ResNet) | BERT | *None* | 68.75 | 44.34 |
| | | CIFAR-100 (VIT) | 63.87 | 7.42 |
| | VIT | *None* | 82.81 | 30.27 |
| | | CIFAR-100 (VIT) | 69.73 | 7.42 |
| CIFAR-10 (VIT) | BERT | *None* | 41.80 | 36.33 |
| | | CIFAR-100 (VIT) | 73.63 | 51.17 |
| | VIT | *None* | 80.86 | 45.90 |
| | | CIFAR-100 (VIT) | 79.88 | 47.66 |
| MNIST (VIT) | BERT | *None* | 98.05 | 92.77 |
| | | CIFAR-10 (VIT) | 96.29 | 90.43 |
| | | CIFAR-100 (VIT) | 97.85 | 89.84 |
| | VIT | *None* | 98.24 | 91.41 |
| | | CIFAR-10 (VIT) | 97.66 | 87.50 |
| | | CIFAR-100 (VIT) | 97.66 | 86.91 |

When designing a defender, minimal tuned parameters and robustness are crucial. Linear, FFN, and Bottleneck, being more flexible with additional tuned parameters during training, tend to bias towards clean data. Conversely, **CeTaD**'s fixed blocks with fewer tuned parameters, exhibit greater robustness, resulting in superior performance. Further exploration on **CeTaD**'s tuned parameters is detailed in Section 4. Evaluating the residual connection's role in **CeTaD**, Table 5 showcases that without this module, both clean and adversarial accuracy degrade significantly, highlighting the crucial role of the residual connection with minimal tuned parameters and one-shot adversarial examples.

**Generalization of CeTaD on Different Attacks**   In reality, deployed service models face various attack methods. To gauge defenders' reliability, we subject them to different attack methods, following default experimental settings. Table 6 showcases **CeTaD**'s adaptability performs well

Table 5: Accuracy performance on the residual connection. *without-res* stands for removing the residual connection.

| Defender | CA(%) | AA(%) |
|---|---|---|
| *None* | 93.75 | 00.00 |
| **CeTaD**-BERT | 68.75 | 44.34 |
| **CeTaD**-BERT-*without-res* | 11.13 | 10.55 |
| **CeTaD**-VIT | 82.81 | 30.27 |
| **CeTaD**-VIT-*without-res* | 12.89 | 12.89 |

Table 6: Accuracy performance against different attack methods. *None* represents no attack method is applied.

| Attack Method | CA(%) | AA(%) |
|---|---|---|
| *None* | 93.75 | - |
| $l_\infty$-PGD | 68.75 | 44.34 |
| $l_\infty$-AutoAttack | 70.70 | 49.41 |
| $l_2$-PGD | 76.17 | 57.03 |
| $l_2$-AutoAttack | 73.44 | 61.33 |

against AutoAttack. We observe that within AutoAttack, only Auto-PGD succeeds; this method is the initial step of the ensemble and singularly overwhelms the victim model. Auto-PGD adjusts its step size automatically, seeking minimal efficient perturbations. However, this pursuit of minimal perturbations may compromise their robustness, allowing for more successful defense strategies. Thus, maintaining a balance between maximum distortion and perturbation effectiveness is critical for generating resilient perturbations. In addition, we include another attack method, StAdvAttack, in Table 3. With a tougher attack method, our method shows good generalization as well.

**Zero-shot Transfer to Different Datasets**   Given the generalization potential of pre-trained models [19, 16, 26], we assess **CeTaD** across varied datasets without re-tuning, named zero-shot transfer. Table 4 indicates that transferring to ResNet on CIFAR-10 from VIT on CIFAR-100 yields lower adversarial accuracy, even worse than random selection. When the target model is VIT, **CeTaD** exhibits improved transfer performance. This sensitivity to victim model structures suggests challenges in direct transfer across different models. Instead, similarity between victim models might aid beneficial transfers between tasks. Notably, the transferred BERT defender achieves higher adversarial accuracy, indicating superior performance of **CeTaD** with diverse prior knowledge.

Further evaluations consider MNIST as the target dataset and CIFAR-10 or CIFAR-100 as the source. Performance remains consistent regardless of the source dataset, suggesting uniform transferable knowledge among these defenders. Shifting focus to more challenging target tasks, Table 7 shows surprising results: defenders tuned on MNIST demonstrate superior adversarial accuracy on CIFAR-100 compared to CIFAR-10. This highlights that transfer from unrelated data might bolster defender robustness. In summary, leveraging transfer gaps enhances defense robustness, potentially empowering defenders across diverse datasets to strengthen performance on individual datasets.

**Effect of Pre-trained Models on CeTaD**   We analyze how two pre-trained models affect **CeTaD**'s performance across MNIST, CIFAR-10, and CIFAR-100 datasets, employing default settings from Section 4. Table 8 highlights the vulnerability of the original service model in the absence of a defender. Despite limited tunable parameters and access to only one-shot adversarial examples, **CeTaD**-equipped models well classify adversarial samples. Notably, **CeTaD** defends both VIT and ResNet on CIFAR-10, demonstrating its adaptability

Table 7: Accuracy performance on zero-shot transfer from bottom to top.

| Defender | Source Data (Source Model) | CA(%) | AA(%) |
|---|---|---|---|
| | *None* | 44.53 | 34.77 |
| BERT | CIFAR-10 (VIT) | 13.87 | 12.89 |
| | MNIST (VIT) | 26.37 | 23.44 |
| | *None* | 52.34 | 30.47 |
| VIT | CIFAR-10 (VIT) | 45.31 | 27.54 |
| | MNIST VIT) | 49.41 | 28.91 |

across diverse victim models. In addition, as shown in Table 9, **CeTaD** could be applied to a larger dataset such as Imagenet-1k. Furthermore, the effective performance of BERT and VIT defenders suggests the potential universality of frozen modules, aligning with prior studies [26, 19].

Overall, defense performance varies based on dataset and defender initialization. MNIST's clear number pixels and consistent backgrounds enable effective defense with both defenders. Conversely, CIFAR-10 and CIFAR-100's diverse scenes pose challenges; tuning introduces bias, impacting clean accuracy. **CeTaD**-VIT defenders excel in clean accuracy, while **CeTaD**-BERT defenders perform better in adversarial scenarios. **CeTaD**-VIT's stability from similar training tasks renders it vulnerable to adversarial perturbations, whereas **CeTaD**-BERT's diverse training complicates clean classification.

Table 8: Accuracy performance of **CeTaD** on different datasets. *None* represents no defense strategy. **TC(mins)** refers to time cost.

| Dataset | Model | Defender | CA(%) | AA(%) | TC |
|---------|-------|----------|-------|-------|-----|
| MNIST | VIT | *None* | 98.83 | 00.78 | 0 |
| | | BERT | 98.05 | 92.77 | 24 |
| | | VIT | 98.24 | 91.41 | 22 |
| CIFAR-10 | ResNet | *None* | 93.75 | 00.00 | 0 |
| | | BERT | 68.75 | 44.34 | 14 |
| | | VIT | 82.81 | 30.27 | 14 |
| | VIT | *None* | 98.05 | 00.00 | 0 |
| | | BERT | 41.80 | 36.33 | 19 |
| | | VIT | 80.86 | 45.90 | 25 |
| CIFAR-100 | VIT | *None* | 91.41 | 00.00 | 0 |
| | | BERT | 44.53 | 34.77 | 32 |
| | | VIT | 52.34 | 30.47 | 28 |

Table 9: Accuracy performance on Imagenet-1k.

| Method | CA(%) | AA(%) |
|--------|-------|-------|
| *None* | 81.64 | 00.00 |
| R&P | 78.71 | 26.56 |
| RN(std=0.1) | 75.98 | 10.16 |
| RN(std=0.2) | 57.42 | 35.55 |
| RN(std=0.3) | 34.18 | 24.41 |
| Linear | 47.66 | 39.45 |
| FFN | 25.20 | 14.84 |
| Bottleneck | 40.63 | 22.85 |
| FD | 52.15 | 27.15 |
| **CeTaD**-GPT-2 | 61.52 | 30.31 |
| **CeTaD**-VIT | 51.17 | 34.38 |
| **CeTaD**-VIT-large | 45.70 | 36.33 |
| **CeTaD**-BERT | 53.32 | 36.91 |
| **CeTaD**-BERT-large | 65.63 | 43.55 |

While humans perceive similarity between clean and adversarial examples, networks struggle, resulting in clean data performance drops due to catastrophic forgetting [14]. Additionally, treating the defender as a prompt generator implies prompts added to examples, guiding the service model to focus on adversarial features, potentially disregarding clean features.

**Discussion on Convergence and Overfitting**  We address two key concerns: 1) Can **CeTaD** effectively adapt to adversarial examples with most parameters frozen and minimal tuning? 2) Given the default one-shot adversarial examples in the training data, is **CeTaD** susceptible to overfitting?

To assess these, we track clean and adversarial accuracy on both training and test data using default experimental settings. However, the limited quantity of training data may limit the expressiveness of accuracy. To gain deeper insights into the training process, we also monitor the training loss on the training data.

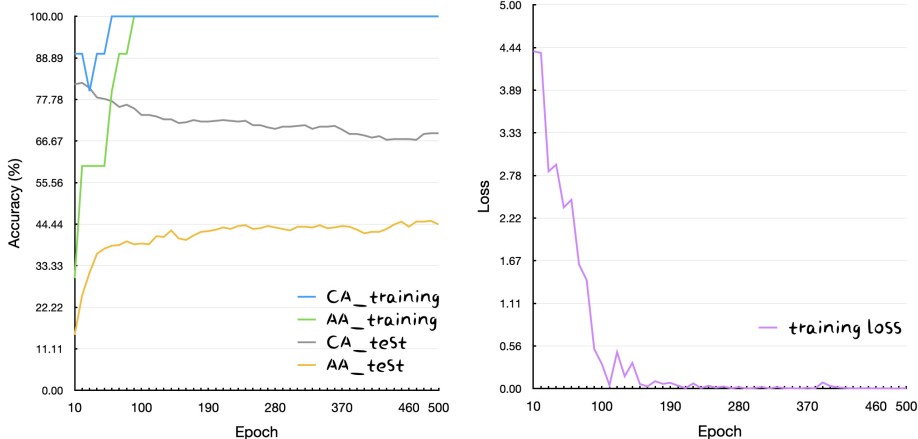

Figure 2: Accuracy and loss over epochs. **Top**: Accuracy curves representing training and test data. *Training* refers to accuracy on training data, while *Test* denotes accuracy on test data. It's notable that clean training data remains unseen during training. **Bottom**: The loss curve depicting training data. Given the consistent 100% accuracy after 90 epochs, this loss curve provides insights into the training process.

Figure 2 illustrates compelling insights. Initially, within 90 epochs, **CeTaD** swiftly reaches 100% adversarial accuracy on training data, showcasing its rapid adaptation, mainly tuning layer norm parameters. Surprisingly, clean accuracy simultaneously reaches 100%, suggesting that training on adversarial examples can unveil reflective feature of clean examples, even when they are not directly presented to the model.

On test data, adversarial accuracy steadily improves, signifying **CeTaD**'s capacity to generalize from a single-shot adversarial dataset. However, this gain comes at the cost of declining clean accuracy. The divergence in distributions and mapping between clean and adversarial data, due to added perturbations, causes a trade-off: aligning with adversarial data benefits but compromises performance on clean data.

In the latter 400 epochs, while maintaining 100% training accuracy, test adversarial accuracy continues a slight ascent, clean accuracy gently declines, and the loss sporadically fluctuates. This pattern suggests that rather than overfitting, **CeTaD** continues to explore and assimilate information from adversarial examples. It is a crucial capability in **RaPiD**, where conventional methods like evaluation or early stopping might not be feasible due to limited training data.

**Role of Pre-trained Initialization and Frozen Parameters** As highlighted in Section 4, the initialization strategies and tuned parameters play a pivotal role in defender performance. Here, we delve into these aspects within **CeTaD**. Given the identical transformer layer structure, the divergence between the BERT and VIT defenders lies primarily in weight initialization. Table 10 illustrates that tuning all parameters generally diminishes clean and adversarial accuracy. Here, the fixed modules in the pre-trained VIT, aligned with image classification, result in a closer mapping relationship between the defender with limited tuning and the victim service, rendering it susceptible to adversarial examples. Contrastingly, comprehensive parameter tuning for VIT fosters a divergence from the original mapping relationship, reinforcing robustness and elevating adversarial accuracy. Notably, the BERT defender excels in adversarial accuracy, while the VIT defender showcases superiority in clean accuracy. Even

Table 10: Performance across various initialization strategies and parameter tuning levels are evaluated. "Random" denotes random initialization of **CeTaD**. The "Tune-All" suffix signifies optimization for all parameters within module **D**, whereas the absence of a suffix indicates the original **CeTaD**.

| Defender | CA(%) | AA(%) |
|---|---|---|
| *None* | 93.75 | 00.00 |
| Random | 52.93 | 42.39 |
| Random-Tune-All | 43.36 | 33.79 |
| BERT | 68.75 | 44.34 |
| BERT-Tune-All | 59.77 | 44.14 |
| VIT | 82.81 | 30.27 |
| VIT-Tune-All | 69.14 | 36.14 |

the randomly initialized defender surpasses the VIT defender in adversarial accuracy. Thus, the VIT-initialized defender appears suboptimal and conservative in comparison.

**Effect of Training Data on CeTaD** The default setup offers only one-shot and unbalanced adversarial examples for swift defense. For example, only 10 adversarial examples sampled randomly are available on CIFAR-10. To investigate how training data variations impact **CeTaD**'s performance, we relax these constraints for assessment. Table 11 highlights that, under default setups, introducing one-shot clean examples as auxiliary data, considering four-shot adversarial examples, or balancing the class distribution in training data significantly improves both clean and adversarial accuracy. Notably, establishing class-balanced data and supplementing clean examples play crucial roles in enhancing CA.

Table 11: Accuracy performance on different training data settings. *1adv* (*1clean*) stands for one-shot adversarial (clean) examples respectively. *Balanced* stands for the class balance.

| Training Data | CA(%) | AA(%) |
|---|---|---|
| *1adv* | 68.75 | 44.34 |
| *1adv-1clean* | 76.76 | 48.24 |
| *4adv* | 70.12 | 50.20 |
| *1adv-Balanced* | 77.34 | 49.02 |

**Continuous Attack**. Similar to adaptive attacks, continuous attacks have the access to the existing system including the defender, which is a very demanding setting in practice. However, our method requires limited tuning on few-shot adversarial samples, which may make continuous defense possible and slightly. Treating each Attack-Then-Defense period as a round, we conduct a pilot evaluation of one-shot continuous rapid defense under the *1adv-1clean* setting mentioned in

Table 12: Performance against continuous attack.

| Round | CA(%) | AA(%) |
|---|---|---|
| – | 93.75 | 00.00 |
| 1 | 73.83 | 55.47 |
| 2 | 78.71 | 64.84 |

Section 4. Results in Table 12 demonstrate that both adversarial and clean accuracy is enhanced through continuous adversarial learning. We foresee the potential expansion of **CeTaD** into an active defender against adaptive attacks in the future.

**Black-Box Attack** Black-box attacks are usually more generalized and robust than white-box methods. We evaluate **CeTaD** on two kinds of black-box attacks, square attack [3] and composite adversarial attack [18], under the default settings. For Square Attack, 5000 queries are applied for randomized perturbation search; for Composite Adversarial Attack (CAA6), semantic perturbations

are combined with scheduled ordering. As shown in Table 13 and 14, **CeTaD** is able to adapt to different black-box attacks by one-shot adversarial fine-tuning.

Table 13: Performance against Square Attack.

| method | CA(%) | AA(%) |
|---|---|---|
| None | 93.75 | 00.00 |
| CeTaD-VIT | 83.20 | 74.02 |
| CeTaD-VIT-large | 81.45 | 75.39 |
| CeTaD-BERT | 82.42 | 79.30 |
| CeTaD-BERT-large | 84.57 | 83.59 |

Table 14: Performance against Composite Adversarial Attack.

| method | CA(%) | AA(%) |
|---|---|---|
| None | 93.75 | 00.00 |
| CeTaD-VIT | 85.74 | 61.52 |
| CeTaD-VIT-large | 69.33 | 52.15 |
| CeTaD-BERT | 78.52 | 65.23 |
| CeTaD-BERT-large | 84.18 | 68.36 |

## 5 Discussion: Limitations and Future Work

In the present scope of **RaPiD**, there remains a notable performance gap for **CeTaD** even without accounting for more potent attacks. End-to-end tuning in **CeTaD** tends to impact clean data performance to a certain extent. Investigating the latent clean data features left in adversarial data might potentially preserve clean accuracy. While this paper focuses solely on image classification, **CeTaD** holds promise for broader application in differentiable systems. Future endeavors aim to assess its performance and generalization across diverse tasks. Additionally, exploring non-differentiable methods like genetic algorithms or reinforcement learning could circumvent differentiability constraints.

Furthermore, the choice of initialization strategy and parameter tuning significantly affects **CeTaD**'s efficacy (Section 4). This study primarily assesses three initialization strategies from standard pre-trained models and explores only parameter tuning for layer norm and full defender parameters. Enhanced strategies for initialization and refined parameter selection through data-driven approaches could bolster performance.

The characteristics of training data are pivotal. While this paper mostly utilizes one-shot imbalanced adversarial examples, Section 4 highlights the potential benefits of class-balanced adversarial examples and their mixture with clean data. Relaxing **RaPiD**'s limitations by structuring a training set with few-shot clean and adversarial examples might optimize performance.

Considering lifelong learning is imperative. The focus on a single attack method in each experiment doesn't account for the reality where service models encounter diverse attack methods. Developing a defender capable of continuous learning to combat new attacks while leveraging past knowledge is essential. By the way, we believe that the defense for deployed models is a complex system. Though we focus on the core (how to defend), there are many other unresolved important problems, such as how to rapidly detect adversarial examples when the attack happens.

In Section 4, surprising outcomes indicate that indirectly related data transfer performs better than related data transfer. This suggests consistency across differing domains, raising questions about aligning modalities based on such consistency. Moreover, exploring transferability across diverse attack methods and various victim models remains open for future exploration. By integrating multiple service models across different tasks and modalities, a relatively universal defender could strengthen its domain robustness.

## 6 Conclusion

Defending operational service models poses challenges, especially with potential difficulties in fine-tuning during attacks, particularly post-pruning. Reinforcing a more resilient model demands extensive computational resources and time. Rapid defense is vital to prevent further damage rather than waiting for adversarial training or model redeployment. Recent methods might lack efficiency in **RaPiD** due to the extensive training data and numerous parameters, causing methodical delays. We introduce **CeTaD**, capitalizing on pre-trained transformer models' broad applicability, harnessing pre-training's capacity to fortify robustness. **CeTaD** excels in clean and adversarial accuracy within constrained resources, minimizing overfitting through minimal parameter adjustments. Evaluations highlight its efficacy across datasets and attacks, probing **CeTaD**'s components via ablation studies. Additionally, exploring data scale and balance effects, transfer tests demonstrate its potential for broader adaptability, possibly reinforcing robustness through transfer gap analysis.

## Acknowledgments and Disclosure of Funding

This work was supported in part by the National Natural Science Foundation of China under Grant 62206205 and 62471371, in part by the Young Talent Fund of Association for Science and Technology in Shaanxi, China under Grant 20230129, in part by the Guangdong High-level Innovation Research Institution Project under Grant 2021B0909050008, and in part by the Guangzhou Key Research and Development Program under Grant 202206030003.

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

## A  Details of Data Preparations

For reproducibility, we illustrate how to prepare data in the experiments.

All datasets are available from Huggingface: MNIST (`https://huggingface.co/datasets/mnist`), CIFAR-10 (`https://huggingface.co/datasets/cifar10`) and CIFAR-100 (`https://huggingface.co/datasets/cifar100`). The library, *Datasets* (`https://github.com/huggingface/datasets`), which includes the methods mentioned below, is utilized for downloading and splitting data.

**N-shot Training Samples.**  First, we split data by class using *filter*. Then, for each category, two methods, *shuffle* with a given seed and *select* for getting the first $n$ samples, are applied in turn. Finally, we mix the selected samples of all classes by *concatenate_datasets* and *shuffle* with the seed.

**512 Fixed Test Samples.**  We apply *shuffle* with the seed and *select* to get the first 512 samples.

In details, data is class-split via *filter*. Each category undergoes *shuffle* and *select* methods for obtaining $n$ samples, followed by *concatenate_datasets* and *shuffle* for mixing all class samples. As per [31], evaluation involves 512 randomly selected images from the test dataset, reducing computational expenses. We apply *shuffle* with the seed and *select* to get the first 512 samples.

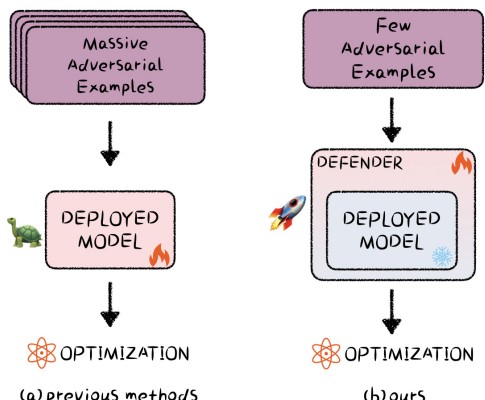

Figure 3: Comparison between previous adversarial training methods and ours: (a) Previous methods heavily rely on vast adversarial examples to tune the original model, demanding significant time and computational resources. (b) In contrast, our approach focuses on tuning only a subset of parameters within the plug-in defender block using limited adversarial examples, enabling swift impact without exhaustive computational demands.

## B  Details of Module Selections inside CeTaD

Module selections are essential for **CeTaD** since only limited parameters are tuned. The embedding and the decoder are vital for feature mapping between the input space and the hidden space. The encoder is significant for perceiving adversarial information and enhancing robustness since it is the only trainable module and bears the most computation in **CeTaD**.

As shown in Figure 1 and illustrated in Section 3, **CeTaD** is flexible as long as the dimensions of the modules match with each other. However, pre-trained weights may help.

For example, we take the embedding from the pre-trained VIT, get the transformer blocks from the pre-trained BERT, VIT or GPT-2, and consider PixelShuffle as the decoder. The modules we used are briefly introduced as follows: BERT ([9]) is a transformer encoder model pre-trained for masked language modeling (MLM) and Next sentence prediction (NSP) on a large corpus of uncased English data (base: `https://huggingface.co/bert-base-uncased`; large: `https://huggingface.co/bert-large-uncased`); VIT ([11]) is a transformer encoder

model pre-trained for image classification on ImageNet-21k at resolution 224x224 (base: `https://huggingface.co/google/vit-base-patch16-224-in21k`; large: `https://huggingface.co/google/vit-large-patch16-224-in21k`); GPT-2 ([33]) is a transformer decoder model pre-trained for causal language modeling (CLM) on a large corpus of English data (124M: `https://huggingface.co/gpt2`); PixelShuffle ([37]) rearranges elements unfolding channels to increase spatial resolution [2].

## C   Details of Optimization

Optimization loops are implemented by PyTorch. To optimize limited parameters and freeze the others, following [26], we set *requires_grad=True* for tunable parameters while *requires_grad=False* for the others. The optimizer is initialized by registering the parameters with *requires_grad=True*. Under the default experimental setup, only layer norm parameters (48 parameter groups, 36864 variables in total) are tuned. We use seed 42 for reported accuracies following [26], each experiment running within 30 minutes on an NVIDIA RTX A5000 GPU.

By the way, the implementation of Lion ([6]), the optimizer which we apply, is available at `https://github.com/lucidrains/lion-pytorch`.

## D   Memory&Latency

We evaluate GPU memory (peak value) and inference time (average value per batch) on the test set under other default settings with the following device configuration: CPU, 14 vCPU Intel(R) Xeon(R) Gold 6330 CPU @ 2.00GHz; GPU, 1 NVIDIA RTX 3090(24GB). As shown in Table 15, our method would not lead to a heavy non-trivial inference overhead. However, with larger pre-trained models, the memory and latency increase accordingly. It is supposed to be a trade-off between defense performance and resource consumption.

Table 15: Accuracy performance on different training data settings. *1adv* (*1clean*) stands for one-shot adversarial (clean) examples respectively. *Balanced* stands for the class balance.

| method | GPU memory (MB) | time (s/batch) |
|---|---|---|
| No-defender | 2186 | 0.07818 |
| CeTaD-BERT | 2638 | 0.08014 |
| CeTaD-BERT-large | 3460 | 0.08496 |

## E   JPEG Compression for Defense

Our evaluation includes naive and training-free defense methods such as Random Noise and R&P. Results show that Random Noise could not balance clean and adversarial accuracy while R&P keeps high clean accuracy but has little effect on adversarial accuracy. We further evaluate that whether an old and simple method, JPEG compression with different quality factors, could work. As shown in Table 16, it is also poor on adversarial accuracy. To conclude, these methods do not work well in the RaPiD scenario, which is more practical yet challenging.

## F   Error Bars

Following [31], we evaluate the accuracy on a fixed subset of 512 images randomly sampled from whole test data to save computational cost. Besides, because of the number of experiments and the page limit, following [26], in the content, we only report the results with one seed (42—*the answer to the ultimate question of life, the universe and everything*). In this section, to show the validity of the results in the content, we additionally repeat two experiments described in Section 4 and Section 4 with another two seeds (41 and 43).

---

[2]In the experiments, *upscale_factor* is always set to 16. Thus, if the scale of the transformer encoder is large, which means the hidden feature is of 1024 dimensions and four channels are given after PixelShuffle, we just ignore the last channel for simplicity.

Table 16: Accuracy performance with JPEG compression under default settings.

| quality factor | CA(%) | AA(%) |
|---|---|---|
| 90 | 90.63 | 01.37 |
| 60 | 88.48 | 09.77 |
| 40 | 86.72 | 10.55 |
| 30 | 86.33 | 09.96 |
| 10 | 82.62 | 08.98 |
| 1 | 71.09 | 06.45 |

In Table 8, Table 17 and Table 18, with a different seed, though the training data and the fixed subset for evaluation vary, leading to accuracy fluctuation, the relative performances of different methods remain the same. Specifically, as illustrated in Section 4, VIT defenders are better at clean accuracy while BERT defenders are likely to outperform at adversarial accuracy. Furthermore, the trends of the corresponding curves in Figure 2, Figure 4 and Figure 5 are similar. It demonstrates that our experiments are both efficient and effective.

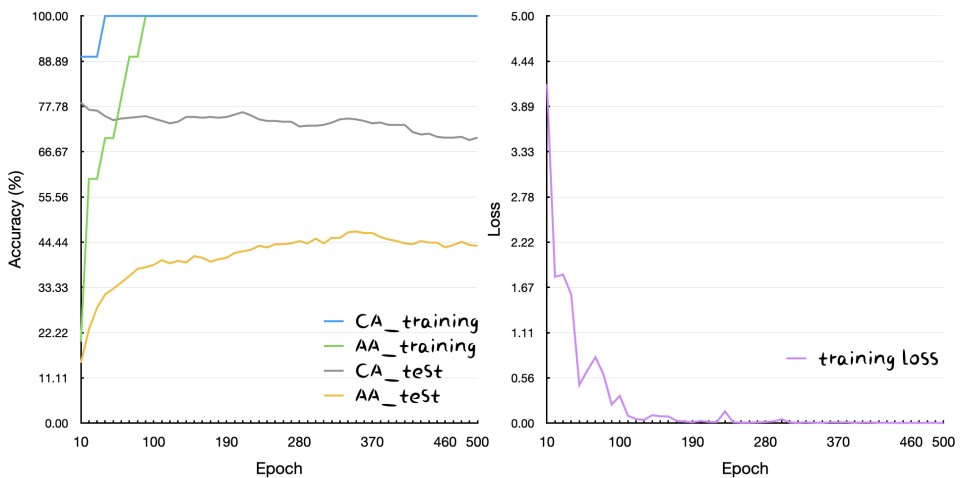

Figure 4: Accuracy and loss vs. epoch with seed 41.

Table 17: Accuracy performance with seed 41.

| Dataset | Model | Defender | CA(%) | AA(%) |
|---|---|---|---|---|
| MNIST | VIT | *None* | 99.02 | 00.59 |
| | | BERT | 97.07 | 90.82 |
| | | VIT | 99.02 | 91.60 |
| CIFAR-10 | ResNet | *None* | 93.95 | 00.00 |
| | | BERT | 70.12 | 43.55 |
| | | VIT | 76.95 | 28.91 |
| | VIT | *None* | 97.85 | 00.00 |
| | | BERT | 35.94 | 31.84 |
| | | VIT | 76.37 | 41.60 |
| CIFAR-100 | VIT | *None* | 91.80 | 00.39 |
| | | BERT | 50.78 | 38.28 |
| | | VIT | 54.30 | 31.45 |

Table 18: Accuracy performance with seed 43.

| Dataset | Model | Defender | CA(%) | AA(%) |
|---|---|---|---|---|
| MNIST | VIT | *None* | 99.22 | 00.59 |
| | | BERT | 98.83 | 93.36 |
| | | VIT | 99.22 | 87.70 |
| CIFAR-10 | ResNet | *None* | 95.51 | 00.00 |
| | | BERT | 73.05 | 44.73 |
| | | VIT | 79.30 | 32.81 |
| | VIT | *None* | 98.05 | 00.00 |
| | | BERT | 69.73 | 53.52 |
| | | VIT | 80.86 | 53.13 |
| CIFAR-100 | VIT | *None* | 94.14 | 00.20 |
| | | BERT | 44.14 | 34.18 |
| | | VIT | 47.07 | 28.32 |

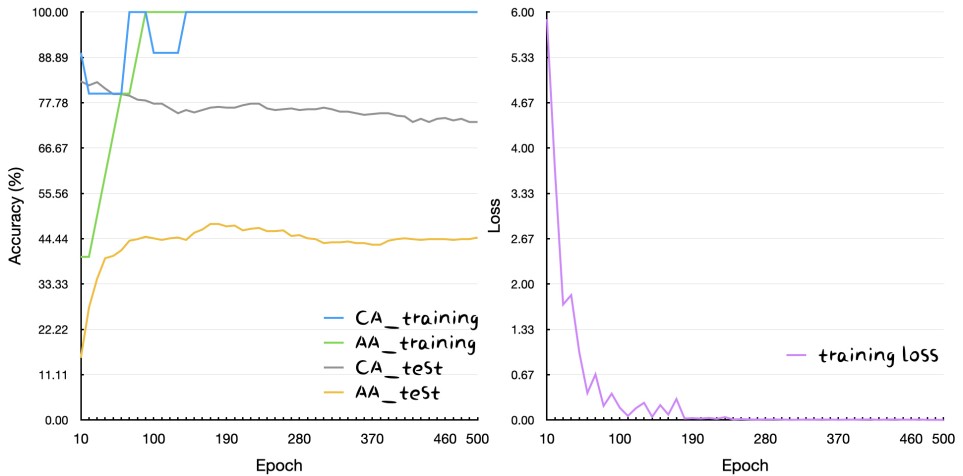

Figure 5: Accuracy and loss vs. epoch with seed 43.

