# OpenReview forum: "Rapid Plug-in Defenders"
_NeurIPS.cc/2024/Conference — NeurIPS 2024 poster_

### Official Review · Reviewer_tz4d · 2024-07-08

**Soundness:** 3
**Presentation:** 3
**Contribution:** 3
**Rating:** 7
**Confidence:** 4

**Summary:**

The paper proposes a method called CeTaD (Considering Pre-trained Transformers as Defenders) for the Rapid Plug-in Defender (RaPiD) problem, which aims to rapidly counter adversarial perturbations without altering the deployed model. The method leverages pre-trained transformer models and fine-tunes only a limited set of parameters using clean and adversarial examples.  The paper evaluates the effectiveness of CeTaD in various scenarios and datasets, and compares it with other baselines. The results show that CeTaD achieves superior performance in both clean and adversarial accuracy compared to the baselines. This paper is easy to follow and well-written.

**Strengths:**

* The proposed CeTaD can rapidly counter adversarial perturbations without altering the deployed model, which addresses an important challenge in the field.
* The CeTaD uses the pre-trained transformer models, which have strong generalization capabilities, and fine-tunes only a limited set of parameters, making it computationally efficient.
* Comprehensive evaluation of CeTaD's effectiveness, transferability, and the impact of different components in various scenarios and datasets are validated.

**Weaknesses:**

* The performance of different l_p bound perturbations should be compared.
* Actually, there are some old defense methods that can also be a plug-in defense, like JPEG compression, which also should be compared.
* Some typo errors, like "Baselines" should be "Baselines."
* The placement of Table 4 and Table 5 are too far from the corresponding content.
* The experiment settings of adaptive attack is not right, in this setting, the plug-in defense is all known to the attacker.

**Questions:**

/NA

**Limitations:**

/NA

---

> ### Author Rebuttal · Authors · 2024-08-07
>
> Thank you for your careful review and overall positive evaluation. Here are our point-by-point responses to your concerns.
>
> **1. l_p settings**
>
> For the experiments in the current paper, we follow the same attack settings as in the related works. We will compare the performance of different ( l_p ) settings in the revised version, just as you suggested. Essentially, with larger ( l_p ), the clean sample becomes more distorted, the attack is stronger, and the defense is more challenging.
>
> (We follow the same attack settings in the related works. We would compare the performance of different l_p settings in the revised version. Basically, with larger l_p, the clean sample would be more distorted, the attack is stronger while the defense is more difficult.)
>
> **2. some old defense methods**
>
> Our evaluation includes naive and training-free defense methods such as Random Noise [30] and R&P [43] (Table 2&3). Random Noise adds zero-mean normal distribution noise; R&P employs random resizing and padding, which is similar to JPEG compression. Results show that Random Noise could not balance clean&adversarial accuracy while R&P keeps high clean accuracy but has little effect on adversarial accuracy. JPEG compression is also poor on adversarial accuracy. Under the setting of Table 2, here's the performance of JPEG compression by changing the quality factor:
>
>
> quality factor| CA(%) | AA(%)
> --- | --- | ---
> 90 | 90.63 | 01.37
> 60 | 88.48 | 09.77
> 40 | 86.72 | 10.55
> 30 | 86.33 | 09.96
> 10 | 82.62 | 08.98
> 1 | 71.09 | 06.45
>
> To conclude, these older methods do not work well in the RaPiD scenario, which is more practical yet challenging.
>
>
> **3. typo errors & table replacement**
>
> We will thoroughly proofread our paper to address all typos and adjust the table placement for better legibility in the revised version.
>
> **4. settings of adaptive attack**
>
> Unlike related works, our method requires limited tuning on few-shot adversarial samples, which may make continuous Attack & Defense possible and slightly different from usual adaptive attacks. In continuous Attack & Defense (Table 12), the plug-in defense in the last round is known to the attacker. We will clarify and emphasize this point in the revised version.

---

> > ### Comment · Reviewer_tz4d · 2024-08-11
> >
> > I am satisfied with the rebuttal and thus keep my rate unchanged.

---

> > > ### Author Response · Authors · 2024-08-11
> > > **Thanks**
> > >
> > > Thank you very much for your support. We will incorporate all of your suggestions into the revised version.

---

### Official Review · Reviewer_ZnXi · 2024-07-11

**Soundness:** 2
**Presentation:** 2
**Contribution:** 2
**Rating:** 4
**Confidence:** 4

**Summary:**

The paper proposes CeTaD, a rapid plug-in defender (RaPiD) for deep neural networks (DNNs) against adversarial attacks. It leverages pre-trained transformer models and fine-tunes minimal parameters (e.g., layer normalization) using few-shot clean and adversarial examples. CeTaD aims to quickly counter adversarial perturbations without modifying the deployed model, addressing the limitations of existing methods that require heavy training or struggle to adapt to different scenarios. Experiments on various datasets and attack methods demonstrate CeTaD's effectiveness, transferability, and generalization capabilities.

**Strengths:**

The paper addresses a practical problem in deep learning security: the need for rapid defense against adversarial attacks without modifying deployed models.

The proposed CeTaD method is computationally efficient, requiring minimal parameter tuning and few-shot examples.

**Weaknesses:**

The discussion had limitations of the paper seems to have made this quite clear: It seems that the effectiveness is still not good enough when we consider stronger attacks. It is somewhat unclear how should we compare this with existing defenses

**Questions:**

How does CeTaD compare to state-of-the-art defense methods in terms of both effectiveness and efficiency?
Can CeTaD effectively defend against stronger adaptive attacks where the attacker has access to the defender?
What are the underlying mechanisms that contribute to CeTaD's effectiveness? A more in-depth analysis would be valuable.

---

> ### Author Rebuttal · Authors · 2024-08-07
>
> Thank you for your review and valuable suggestions. Here is our response to your concerns.
>
> **1. effectiveness and efficiency**
>
> This paper compares methods capable of implementing RaPiD (Rapid Plug-in Defender) (see L130-L134). As a very promising line of technique, there are indeed existing works in purification and denoising methods, which we introduce in our paper (e.g., L40 to L48). That being said, our paper tackles more challenging scenarios with innovative technical developments. Specifically, we overcome the hurdle that existing defenses are not sufficiently capable of rapidly defending with limited data or adapting to different application scenarios. Our proposed method features the following novel designs: 1) the pre-trained model is applied for robustness; 2) limited tuning with only few-shot adversarial samples is utilized for scenario adaptation; 3) most parameters are fixed to maintain clean accuracy. These new technical designs yield better performance in robustness and generalization when compared to exisiting methods. Additionally, our method has the potential to support many extensions for better future work, as elaborated in the Discussion (Sec.5 Page 9).
>
> **2. stronger attacks**
>
> We evaluate our method on different attacks including PGD, AutoAttack and StAdvAttack. Besides, in the continuous Attack&Defense evaluation (L354-L363; Table 12), we simulate the situation that the defender is also leaked. The result shows that both CA and AA are continuously improved, which may support that life-long learning is a potential direction. Due to the limited time of the rebuttal period, we would try to extend more additional evaluations on black-box attacks and universal perturbations in the final version.
>
>
> **3. underlying mechanism**
>
> We have made comprehensive evaluation and analysis on the underlying mechanism of our method.
>
> (L245-251) We do ablation studies on the structure. Linear, FFN, and Bottleneck, being more flexible with additional tuned parameters during training, tend to much bias towards clean data. Conversely, CeTaD’s fixed blocks with fewer tuned parameters, exhibit greater robustness, resulting in superior performance. limited tuning would avoid completely damaging the original capability while adapting the whole framework to the scenario.
>
> (L290-296; L326-L345) We do evaluations on different parameter initializations and frozen parameters. Randomly initialized parameters do not work well and tuning all parameters would ruin the clean accuracy. CeTaD-VIT defenders excel in clean accuracy, while CeTaD-BERT defenders perform better in adversarial scenarios. CeTaD-VIT’s capability from similar training tasks renders it vulnerable to adversarial perturbations, whereas CeTaD-BERT’s diverse training complicates clean classification. Totally, introducing fixed&unrelated structure would improve the robustness.

---

> > ### Author Response · Authors · 2024-08-14
> >
> > Thank you for your time and effort in reviewing our work, and we really appreciate your support.
> >
> > There is only 1 day left to the rebuttal deadline, and we would like to know whether our responses can successfully address your concerns. Please also let us know if you have other concerns, and we are more than happy to address them.
> >
> > Best Wishes
> >
> > Authors

---

> > ### Author Response · Authors · 2024-08-14
> > **New results on Stronger attacks**
> >
> > We will incorporate all of your suggestions into the revised version.
> >
> > **1. memory&latency**
> >
> > We evaluate GPU memory (peak value) and inference time (average value per batch) on the test set under other default settings with the following device configuration:
> >
> > - CPU: 14 vCPU Intel(R) Xeon(R) Gold 6330 CPU @ 2.00GHz
> > - GPU: 1 NVIDIA RTX 3090(24GB)
> >
> > Here is the result:
> >
> > | method | GPU memory (MB) | time (s/batch) |
> > | --- | --- | --- |
> > | No-defender | 2186 | 0.07818 |
> > | CeTaD-BERT | 2638 | 0.08014 |
> > | CeTaD-BERT-large | 3460 | 0.08496 |
> >
> > With larger pre-trained models, the memory and latency increase accordingly. It is supposed to be a trade-off between defense performance and resource consumption.
> >
> > **2. other attacks**
> >
> > **black-box attack**
> >
> > Additionally, we follow your suggestion to evaluate our method with a pure black-box attack, Square Attack [1] (n_queries=5000), under the default settings. Here is the result on CIFAR-10 indicating that our method is still effective against Square Attack:
> >
> > Type of attack |method |CA(%)|AA(%)
> > --- |--- | --- | ---
> > SquareAttack |None| 93.75 |00.00
> > ||CeTaD-VIT|83.20|74.02
> > ||CeTaD-VIT-large|81.45|75.39
> > ||CeTaD-BERT|82.42|79.30
> > ||CeTaD-BERT-large|84.57 |83.59
> > L_inf-PGD|CeTaD-VIT| 82.81|30.27
> > ||CeTaD-VIT-large |71.68 |44.14
> > ||CeTaD-BERT |68.75 |44.34
> > ||CeTaD-BERT-large |66.02 |48.83
> >
> > Our method, CeTaD, demonstrates superior defense effectiveness against black-box attacks compared to L_{\infty}-PGD. Our approach fine-tunes CeTaD using adversarial examples generated from attacks to detect attack patterns and perform defense, without requiring any information about the attack strategies. As a result, it is not limited to any specific type of attack.
> >
> > **semantic attacks & composite attacks**
> >
> > We also tested CeTaD’s performance against semantic attacks and composite attacks. The experimental setup was consistent with [2], using WideResNet34 trained on CIFAR-10. CeTaD was trained using a 1-adv-1-clean data setting, with significantly fewer samples than the method in [2], and with a much shorter training time.
> >
> > Type of attack |method |CA(%)|AA(%)
> > --- |--- | --- | ---
> > Semantic attacks |CeTaD-BERT-large|81.23 |64.59
> > Semantic attacks |R&P|91.2 |3.84
> >
> > However, our experimental results show that our method achieves good performance against semantic attacks.
> >
> > Additionally, we also test a composite attack, Composite Adversarial Attack [2] (CAA6: enabled_attack=(0,1,2,3,4,5); order_schedule="scheduled"), under the default settings. The results indicating that our framework is able to adapt to different scenarios by one-shot adversarial fine-tuning.
> >
> > | method | CA(%) | AA(%) |
> > | --- | --- | --- |
> > | None | 93.75 | 00.00 |
> > | CeTaD-VIT | 85.74 | 61.52 |
> > | CeTaD-VIT-large | 69.33 | 52.15 |
> > | CeTaD-BERT | 78.52 | 65.23 |
> > | CeTaD-BERT-large | 84.18 | 68.36 |
> >
> > [1] Square Attack: a query-efficient black-box adversarial attack via random search
> >
> > [2] Towards Compositional Adversarial Robustness: Generalizing Adversarial Training to Composite Semantic Perturbations

---

### Official Review · Reviewer_Dpo5 · 2024-07-17

**Soundness:** 3
**Presentation:** 2
**Contribution:** 3
**Rating:** 4
**Confidence:** 4

**Summary:**

This submission studies few-shot tuning-based purification for adversarial defense, especially leveraging pre-trained transformers. By only tuning the normalization layers with few training examples, the proposed defense achieves decent accuracy and robustness.

**Strengths:**

1. The problem setting is novel and of practical value. Existing adversarial defense requires much computing and data and lacks generalization. The RaPiD problem is of value.

2. The proposed method has strong empirical performance. By leveraging existing pre-trained transformers and ViT as the purifier and a few examples to fine-tune the layer norm parameters, the proposed defense has strong accuracy and robustness supported by comprehensive experimental evaluation.

**Weaknesses:**

1. My major concern is the unclear description in both the method description and experimental evaluation (sometimes a bit messy). To name a few:

(1) How does the decoder in Figure 1 work? In my understanding, the decoder should be able to restore the potentially perturbed image. However, existing ViT and language models seem not to be capable of doing so. How does PixelShuffle for superresolution work as a decoder? In my understanding, it should take low-resolution image input which is not provided by the encoder.

(2) What do CA and AA mean? I infer that they are clean accuracy and attacked accuracy in all tables but no place to verify.

(3) When measuring AA, what is the specific attack used? PGD, AutoAttack, and StAdvAttack are mentioned but which one is actually used? Moreover, when using the attack, does the gradient flow go backward through the plug-in defender? When compared with other baselines, this is the required adaptive attack and when compared with other baselines, this defense-then-attack number should be used for fair comparison.

(4) What is StAdv attack? Any setup detail?

(5) What is the dataset in Table 5?

2. The proposed method introduces a large model as a pre-purifier which adds a non-trivial inference overhead. More elaboration on this overhead should be discussed. In what case this overhead can be neglected?

Given these many unclear points, I lean towards rejection.

Minor:
1. Table 1: better to mention method names rather than just numbered brackets for compared approaches.
2. Line 112: hallucinates -> hallucinate

**Questions:**

Questions are listed in "Weaknesses" section. The most important one is how the proposed method works given the misaligned encoder and decoder module.

**Limitations:**

Yes, limitations are discussed in Section 5.

---

> ### Author Rebuttal · Authors · 2024-08-07
>
> Thank you for your careful review and valuable comments. We provide point-to-point clarifications on your concerns below and will incoporate them in the revised version.
>
> **1. framework mechanism**
>
> The decoder does not need to restore the input. Encoder-decoder is a general framework. Usually, an encoder maps the data of source space into a latent feature, and a decoder maps the latent feature into the target space. For some methods to extract feature, source and target space are the same and the target of decoder is to restore the input. For others, they are different. For example, for a text translation model, source space may be the original language while target space is the target language.
>
> In our method, the decoder maps latent feature into the space where the original image classification model could correctly work. For efficiency and robustness, we do not want to train a decoder from scratch. Thus, we take PixelShuffle as a simple decoder to adjust channel and size. You could consider PixelShuffle as a fixed&untrainable decoder and we just tuning the encoder to adapt the framework.
>
> **2. CA & AA**
>
> CA is the accuracy on clean data and AA is the accuracy on adversarial data, which can be found in Experimental Setup (L210). We will highlight their definitions to make them clearer.
>
> **3. attack settings**
>
> As described in Experimental Setup (L208), the default attack is PGD. We also evaluate our method on AutoAttack (Table 5) and StAdvAttack (Table 3). In the continuous Attack&Defense evaluation (L354-L363; Table 12), we simulate the situation that the defender is also leaked. The result shows that both CA and AA are continuously improved, which may support that life-long learning is a potential direction.
>
> **4. StAdvAttack**
>
> We illustrate all attack methods for evaluation in Experimental Setup (L199-L201). StAdvAttack is a strong attack proposed in [1].
>
> [1] Spatially transformed adversarial examples
>
> **5. the dataset in Table 5**
>
> We illustrate default settings in Experimental Setup (L208-210). the default dataset is CIFAR-10.
>
> **6. non-trivial inference overhead**
>
> Totally, we agree that the trade-off is important in real-time applications. However, no pains no gains. For RaPiD problem, our method needs much less tuning overhead while outperforms other methods on robustness and generalization. In practical, we could flexibly choose the pre-trained model to balance performance and inference overhead. Besides, there are many speed-up techniques for inference, such as quantization, pruning, and distillation.
>
> **7. Minor**
>
> Thank you for your careful proofreading, which greatly helps us improve our paper. We will address the mentioned details in the revised version.

---

> > ### Author Response · Authors · 2024-08-14
> >
> > Thank you for your time and effort in reviewing our work, and we really appreciate your support.
> >
> > There is only 1 day left to the rebuttal deadline, and we would like to know whether our responses can successfully address your concerns. Please also let us know if you have other concerns, and we are more than happy to address them.
> >
> > Best Wishes
> >
> > Authors

---

> > ### Comment · Reviewer_Dpo5 · 2024-08-14
> >
> > Thanks for the response! Most of my concerns are resolved. I adjusted the score accordingly. However, here are follow-up questions:
> >
> > > 1. framework mechanism
> >
> > Now I understand how the decoder works here. Then, my questions are:
> >
> > (1) For pure language model encoders, such as BERT and GPT-2, how do they perceive images? If you use an image encoder, what is it and are its weights fixed?
> >
> > (2) For decoder-only transformers like GPT-2, how to attach a decoder like PixelShuffle next to it? GPT-2 does not generate embeddings directly.
> >
> > > 6. non-trivial inference overhead
> >
> > It would be great to quantitatively study the overhead across different datasets. For example, I would expect the overhead is too high for MNIST and may be tolerable for ImageNet.

---

> ### Author Response · Authors · 2024-08-14
>
> I wish to express my sincere appreciation for the invaluable assistance and dedication you have provided during the review process of my paper. Your insightful comments and thoughtful suggestions have significantly contributed to the development and refinement of my work. Your efforts have been of immense help in enhancing the quality and clarity of my research. Thank you for your generous support. It is greatly appreciated. We will incorporate all of your suggestions into the revised version.
>
> **1.framework mechanism**
>
> 1) As illustrated in L163-L164 and Figure 1, an image would be processed by the trained embedding from VIT, resulting in the sequence embedding which could be processed by the encoder model.
> Besides, as shown in Figure 1, most parameters are fixed for generalization and robustbess. Only layernorm is tunable. Furthermore, we do additional experiment on tunable parameters in Table 10 for evaluation.
>
> 2) The decoder-based transformer model is similar to the encoder-based one, the difference is the attention mask. Thus, we could apply GPT-2 too. However, causal attention could not well extract needed feature, as shown in Table 2.
>
> **2. non-trivial inference overhead**
>
> We evaluate GPU memory (peak value) and inference time (average value per batch) on the test set under other default settings (cifar10) with the following device configuration:
>
> - CPU: 14 vCPU Intel(R) Xeon(R) Gold 6330 CPU @ 2.00GHz
>
> - GPU: 1 NVIDIA RTX 3090(24GB)
>
> Here is the result:
>
> | method | GPU memory (MB) | time (s/batch) |
> | --- | --- | --- |
> | No-defender | 2186 | 0.07818 |
> | CeTaD-BERT | 2638 | 0.08014 |
> | CeTaD-BERT-large | 3460 | 0.08496 |
>
> Totally, the defender does not need non-trivial inference overhead. However, with larger pre-trained models, the memory and latency increase accordingly. It is supposed to be a trade-off between defense performance and resource consumption. In practical, the developer could choose the suitable pre-trained model for their device and scenario.
>
> **3. other attacks**
>
> **black-box attack**
>
> Additionally, we also evaluate our method with a pure black-box attack, Square Attack [1] (n_queries=5000), under the default settings. Here is the result on CIFAR-10 indicating that our method is still effective against Square Attack:
>
> Type of attack |method |CA(%)|AA(%)
> --- |--- | --- | ---
> SquareAttack |None| 93.75 |00.00
> ||CeTaD-VIT|83.20|74.02
> ||CeTaD-VIT-large|81.45|75.39
> ||CeTaD-BERT|82.42|79.30
> ||CeTaD-BERT-large|84.57 |83.59
> L_inf-PGD|CeTaD-VIT| 82.81|30.27
> ||CeTaD-VIT-large |71.68 |44.14
> ||CeTaD-BERT |68.75 |44.34
> ||CeTaD-BERT-large |66.02 |48.83
>
> Our method, CeTaD, demonstrates superior defense effectiveness against black-box attacks compared to L_{\infty}-PGD. Our approach fine-tunes CeTaD using adversarial examples generated from attacks to detect attack patterns and perform defense, without requiring any information about the attack strategies. As a result, it is not limited to any specific type of attack.
>
> **semantic attacks & composite attacks**
>
> We also tested CeTaD’s performance against semantic attacks and composite attacks. The experimental setup was consistent with [2], using WideResNet34 trained on CIFAR-10. CeTaD was trained using a 1-adv-1-clean data setting, with significantly fewer samples than the method in [2], and with a much shorter training time.
>
> Type of attack |method |CA(%)|AA(%)
> --- |--- | --- | ---
> Semantic attacks |CeTaD-BERT-large|81.23 |64.59
> Semantic attacks |R&P|91.2 |3.84
>
> However, our experimental results show that our method achieves good performance against semantic attacks.
>
> Additionally, we also test a composite attack, Composite Adversarial Attack [2] (CAA6: enabled_attack=(0,1,2,3,4,5); order_schedule="scheduled"), under the default settings. The results indicating that our framework is able to adapt to different scenarios by one-shot adversarial fine-tuning.
>
> | method | CA(%) | AA(%) |
> | --- | --- | --- |
> | None | 93.75 | 00.00 |
> | CeTaD-VIT | 85.74 | 61.52 |
> | CeTaD-VIT-large | 69.33 | 52.15 |
> | CeTaD-BERT | 78.52 | 65.23 |
> | CeTaD-BERT-large | 84.18 | 68.36 |
>
> [1] Square Attack: a query-efficient black-box adversarial attack via random search
>
> [2] Towards Compositional Adversarial Robustness: Generalizing Adversarial Training to Composite Semantic Perturbations

---

> > ### Comment · Reviewer_Dpo5 · 2024-08-14
> >
> > Thanks for your effort in conducting additional evaluations to support the research!
> >
> > 1. framework mechanism
> >
> > Got it. Now I understand what vision embeddings are used. But for the decoder-only transformer model, how to extract the features? Is it the output activation per token before the next-token predictor hand? If so, how are the activations concatenated to form a single feature for PixelShuffle?

---

> ### Author Response · Authors · 2024-08-14
>
> 1.  framework mechanism
> ​
>
> ​We take the output of the last encoder layer, which is in the shape of (batch_size, patch/sequence_size, hidden_size). Then PixelShuffle rearranges (you could consider it as unfolding channels while increasing spatial resolution) the elements of the last two dimension into the shape of the images (batch_size,  H, W). We will incorporate and clarify this information into the revised version.

---

### Official Review · Reviewer_tqbT · 2024-07-29

**Soundness:** 3
**Presentation:** 3
**Contribution:** 3
**Rating:** 5
**Confidence:** 4

**Summary:**

This paper introduces an approach for defending machine learning models against adversarial attacks using transformer-based vision models. The proposed method, Continuous Transfer and Defense (CeTaD), leverages pre-trained transformers as defenders to provide rapid and effective adversarial protection. The approach is evaluated on various datasets, including MNIST, CIFAR-10, CIFAR-100, and ImageNet-1K, to demonstrate its adaptability and effectiveness across different scenarios. However, the approach leads to some impact on classification accuracy.

**Strengths:**

1. This paper provides extensive empirical evaluations across multiple datasets and models, showcasing the proposed method's robustness and versatility.
2. The study on transferability of defenders across different datasets and models is insightful, suggesting potential for practical deployment in diverse applications.
3. The discussion on continuous learning and adaptation against new attacks highlights the method's forward-thinking approach to evolving security challenges.

**Weaknesses:**

1. There are many works using purification or denoising methods in this field. Thus, the proposed method is not novel enough.
2. The paper acknowledges that end-to-end tuning in CeTaD can impact clean data performance, which might be a significant drawback in certain applications​.
3. While the paper evaluates the method on several datasets, the focus remains primarily on image classification tasks. Broader evaluations across other types of tasks and data modalities are needed to fully establish its generalizability.
4. CeTaD's effectiveness relies heavily on the availability and quality of pre-trained models, which might not always be feasible in every scenario.
5. The paper lacks a comprehensive discussion of related works in the field, especially those that use denoising layers and plug-and-play mechanisms.

**Questions:**

1. What are the specific computational requirements for deploying CeTaD in a real-time environment (e.g., memory, latency, efficiency), and how do these requirements scale with the size of the dataset?
2. How does the method handle adversarial examples that are generated using techniques not covered in the evaluation?
3. How does the model perform on black-box attacks or universal perturbations?

**Limitations:**

1. The paper notes a notable performance gap for CeTaD, particularly in terms of clean data accuracy, when applying end-to-end tuning.
2. While the paper discusses the potential for scaling the method to larger datasets, the actual computational efficiency and scalability remain areas for further investigation​.
3. The current focus is on a single attack method per experiment, which does not fully reflect the diversity of attack methods (e.g., semantic attacks, composite attacks, multiple-threat attacks [1, 2]) encountered in real-world scenarios.

[1] Hosseini and Poovendran. Semantic adversarial examples.

[2] Hsiung et al. Towards compositional adversarial robustness: Generalizing adversarial training to composite semantic perturbations.

---

> ### Author Rebuttal · Authors · 2024-08-07
>
> Thanks for your careful review and insightful comments. We provide a point-to-point response to your concerns.
>
> **1. related methods & novelty**
>
> As a very promising line of technique, there are indeed existing works in purification and denoising methods, which we introduce in our paper (e.g., L40-L48). That being said, our paper tackles a more challenging scenario and provides innovative technical developments. Specifically, we overcome the hurdle that existing defenses are not sufficiently capable of rapidly defending with limited data or adapting to different application scenarios. Our proposed method features the following novel designs: 1) the pre-trained model is applied for robustness; 2) limited tuning with only few-shot adversarial samples is utilized for rapid scenario adaptation; 3) most parameters are fixed to maintain clean accuracy. These new technical designs yield better performance in robustness and generalization when compared to existing methods. Additionally, our method has the potential to support many extensions for better future work, as elaborated in the Discussion (L365-L392). Thus, we believe our proposed method is contributive enough.
>
>
> **2. clean data performance**
>
> We would like to clarify that it is an inevitable tradeoff between generalization performance and robustness enhancement as widely acknowledged in literature, which explains the decreased accuracy on clean samples. Our method actually achieves a better tradeoff under a more challenging problem setting. That is, with only limited adversarial data for tuning, our method outperforms other possible methods in balancing both clean and adversarial accuracy (Table 2).
>
> Besides, our method shows the capability to gradually recover the performance of clean data with more samples for training (L346-L353; Table 11) or continous Attack&Defense (L354-L363; Table 12).
>
> **3. other tasks and data modalities**
>
> Thank you very much for your suggestions.
>
> We comprehensively evaluate and analyze our method on different image classification datasets, ensuring the extensiveness of our empirical validation. Additionally, we choose to focus on attacks against the image classification task because they present one of the most significant threats due to the versatile and advanced capabilities of these attacks nowadays, which have already raised significant challenges for the design of defense mechanisms. Therefore, although we agree that it is a valuable future direction to study defenses against attacks on other tasks and domains, it does not diminish the significance of our current work. We will provide more discussions about the possibilities for large-scale, multi-task, and multi-modality applications in future work.
>
>
> **4. dependence on high-quality pre-trained models in the specific scenario**
>
> It is true that some methods rely on high-quality pre-trained models on the specific scenario (such as DiffPure; L45-L48). However, our method does not need a customized pre-trained model for the scenario. In fact, we find an interesting phenomenon: applying BERT (model pre-trained for text) is better than VIT (model pre-trained for image) for the image classification task (Table 2). We think uncustomized parameters matter for robustness here.
>
> **5. requirements for real-time environment**
>
> For our method, the original model is fixed. Only few-shot samples and limited training are needed. Compared with other methods, ours is neither time-consuming nor computationally expensive. In a specific scenario, detailed requirements (e.g., memory, latency, efficiency) rely on the selected pre-trained model, the quantity of tuning samples, available computational resources, inference settings, and so on.
>
> **6. attacks not covered in the evaluation**
>
> Our method is a general framework that does not restrict the type of attacks. For different attacks, it only requires a small number of adversarial samples without any additional operations. Following related methods, such as [1], we evaluate our method against common attacks, including PGD, AutoAttack, and StAdvAttack. These white-box attacks are stronger than black-box attacks or universal perturbations. The results demonstrate the effectiveness and robustness of our method against these white-box attacks. Moreover, due to the time constraints of rebuttal, we would try to extend more additional evaluations on other attacks in the final version.
>
> [1] Diffusion models for adversarial purification, ICML-22.

---

> > ### Comment · Reviewer_tqbT · 2024-08-10
> >
> > Thanks to the author for the rebuttal. I carefully read the author's rebuttal as well as other review comments. Nonetheless, it appears that some aspects of the weaknesses and questions identified in the initial review have not been addressed in the rebuttal, leaving some pieces of my concerns unresolved.
> >
> > First, how the memory and latency scale with the size of the dataset/model remains unclear. Can the authors provide concrete examples to demonstrate this? Second, how does CeTaD handle black-boxed adversarial examples or other unforeseen attacks (e.g., semantic attacks & composite attacks)? This question also remains unanswered. Although white-box attacks are stronger than black-box attacks or universal perturbations, in terms of defense, white-box attacks are easier to defend than black-box attacks. The author's rebuttal does not really convenience me.
> >
> > The paper should conduct more thorough evaluations and discussions on the points mentioned above. Therefore, I decided to keep my score.

---

> > > ### Author Response · Authors · 2024-08-14
> > >
> > > Thank you for your time and effort in reviewing our work, and we really appreciate your support.
> > >
> > > There is only 1 day left to the rebuttal deadline, and we would like to know whether our responses can successfully address your concerns. Please also let us know if you have other concerns, and we are more than happy to address them.
> > >
> > > Best Wishes,
> > >
> > > Authors

---

> ### Author Response · Authors · 2024-08-11
> **New Experimental Results**
>
> Thanks for your further comments. We provide a point-to-point response to your concerns. We will incorporate all of your suggestions into the revised version.
>
> **1. memory&latency**
>
> We evaluate GPU memory (peak value) and inference time (average value per batch) on the test set under other default settings with the following device configuration:
>
> - CPU: 14 vCPU Intel(R) Xeon(R) Gold 6330 CPU @ 2.00GHz
> - GPU: 1 NVIDIA RTX 3090(24GB)
>
> Here is the result:
>
> | method | GPU memory (MB) | time (s/batch) |
> | --- | --- | --- |
> | No-defender | 2186 | 0.07818 |
> | CeTaD-BERT | 2638 | 0.08014 |
> | CeTaD-BERT-large | 3460 | 0.08496 |
>
> With larger pre-trained models, the memory and latency increase accordingly. It is supposed to be a trade-off between defense performance and resource consumption.
>
> **2. other attacks**
>
> **black-box attack**
>
> Additionally, we follow your suggestion to evaluate our method with a pure black-box attack, Square Attack [1] (n_queries=5000), under the default settings. Here is the result on CIFAR-10 indicating that our method is still effective against Square Attack:
>
> Type of attack |method |CA(%)|AA(%)
> --- |--- | --- | ---
> SquareAttack |None| 93.75 |00.00
> ||CeTaD-VIT|83.20|74.02
> ||CeTaD-VIT-large|81.45|75.39
> ||CeTaD-BERT|82.42|79.30
> ||CeTaD-BERT-large|84.57 |83.59
> L_inf-PGD|CeTaD-VIT| 82.81|30.27
> ||CeTaD-VIT-large |71.68 |44.14
> ||CeTaD-BERT |68.75 |44.34
> ||CeTaD-BERT-large |66.02 |48.83
>
> Our method, CeTaD, demonstrates superior defense effectiveness against black-box attacks compared to L_{\infty}-PGD. Our approach fine-tunes CeTaD using adversarial examples generated from attacks to detect attack patterns and perform defense, without requiring any information about the attack strategies. As a result, it is not limited to any specific type of attack.
>
> **semantic attacks & composite attacks**
>
> We also tested CeTaD’s performance against semantic attacks and composite attacks. The experimental setup was consistent with [2], using WideResNet34 trained on CIFAR-10. CeTaD was trained using a 1-adv-1-clean data setting, with significantly fewer samples than the method in [2], and with a much shorter training time.
>
> Type of attack |method |CA(%)|AA(%)
> --- |--- | --- | ---
> Semantic attacks |CeTaD-BERT-large|81.23 |64.59
> Semantic attacks |R&P|91.2 |3.84
>
> However, our experimental results show that our method achieves good performance against semantic attacks.
>
> Additionally, we also test a composite attack, Composite Adversarial Attack [2] (CAA6: enabled_attack=(0,1,2,3,4,5); order_schedule="scheduled"), under the default settings. The results indicating that our framework is able to adapt to different scenarios by one-shot adversarial fine-tuning.
>
> | method | CA(%) | AA(%) |
> | --- | --- | --- |
> | None | 93.75 | 00.00 |
> | CeTaD-VIT | 85.74 | 61.52 |
> | CeTaD-VIT-large | 69.33 | 52.15 |
> | CeTaD-BERT | 78.52 | 65.23 |
> | CeTaD-BERT-large | 84.18 | 68.36 |
>
> [1] Square Attack: a query-efficient black-box adversarial attack via random search
>
> [2] Towards Compositional Adversarial Robustness: Generalizing Adversarial Training to Composite Semantic Perturbations

---

> > ### Comment · Reviewer_tqbT · 2024-08-14
> >
> > Thanks to the author for the experimental results. My main concerns have been addressed, and I believe these results can further demonstrate the applicability of the proposed approach, especially in defending against unseen attacks. I recommend that the authors incorporate these results into the revision. I have updated my scores accordingly.

---

> > > ### Author Response · Authors · 2024-08-14
> > >
> > > I wish to express my sincere appreciation for the invaluable assistance and dedication you have provided during the review process of my paper. Your insightful comments and thoughtful suggestions have significantly contributed to the development and refinement of my work. Your efforts have been of immense help in enhancing the quality and clarity of my research. Thank you for your generous support. It is greatly appreciated. We will incorporate all of your suggestions into the revised version.

---

### Decision · Program_Chairs · 2024-09-25

**Decision:**

Accept (poster)

**Comment:**

The recommendation is based on the reviewers' comments, the area chair's evaluation, and the author-reviewer discussion.

While all reviewers find the studied setting useful and the results provide new insights, in the initial round, reviewers raised several valid questions on the scope and empirical validation. The authors' rebuttal was quite successful and addressed the major concerns of all reviewers. In the final discussion phase, all reviewers suggest that the new results (especially those presented during rebuttal) must be included in the final paper to reach the bar of the publication, which requires a major revision (this will be discussed with SAC). If this paper is to be accepted, we strongly suggest the authors include the new results and improve the presentation for the final version.